# The Impact of Financial Literacy on Household Health Investment: Empirical Evidence from China

**DOI:** 10.3390/ijerph20032229

**Published:** 2023-01-26

**Authors:** Xiao Ling, Luanfeng Wang, Yuxi Pan, Yanchao Feng

**Affiliations:** 1Business School, Hubei University, Wuhan 430062, China; 2Business School, Zhengzhou University, Zhengzhou 450001, China

**Keywords:** financial literacy, household health investment, health literacy, household finance

## Abstract

Based on the 2019 China Household Finance Survey (CHFS) data, this paper used factor analysis to measure the level of financial literacy of surveyed householders and used the Probit model and the negative binomial model to test the impact of financial literacy (*FL*) on household health investment (*HHI*). The results show that: (1) *FL* is an essential influencing factor in increasing participation in *HHI*, and householders with a higher level of *FL* are also more willing to pay for diversified investments. (2) We split the *FL* level from the two dimensions of knowledge and ability. We found that the primary *FL* (including financial knowledge, computing ability, and correct recognition of investment product risk) plays a more critical role in the investment decision process. (3) When information sources, health knowledge, and family income are used as mediating variables, *FL* can influence the decisions of *HHI* in three ways: expanding information sources, enriching health knowledge, and alleviating income constraints. (4) By analyzing the heterogeneity of household heads in different regions and with different personal characteristics, we found that the medical level of the household location and the life and work experience of the householders played a moderating role.

## 1. Introduction

The health problems of Chinese residents are becoming increasingly prominent due to various factors such as urbanization, food structure, and climate change [1,2]. Chronic diseases such as hypertension and diabetes are increasingly seen in young people. “Geriatric diseases” are gradually showing a trend of lower age, the age structure of patients is changing, and the growth rate of chronic disease groups remains high, which brings a substantial economic burden to the healthy life of residents and national health care. In 2020, the prevalence of chronic diseases in China’s physically examined population reached 24% [3], and the number of deaths caused by chronic diseases accounted for 88.5% of annual deaths in China and also generated more than 90% of the national economic disease burden. To cater to the health protection demands of Chinese residents, the government has issued guiding documents such as the “Health China 2030 Planning Outline” and “Health China Action (2019–2030)” to integrate residents’ health as a national strategy into various policies. The growth rate of per capita health care expenditure reached 22.7% between 2010 and 2019, higher than the growth rate of per capita disposable income of 16.1%, and the compound annual growth rate of a commercial health insurance premium income was as high as 29.8% [4,5]. At the same time, residents’ awareness of health protection has increased significantly.

Along with the rapid development of China’s economy and education, more and more families are looking for comprehensive health and financial protection measures to actively cope with health risks brought by aging, environmental pollution, and public health events. The COVID-19 outbreak has also stimulated residents’ demand for health coverage. According to the survey data, residents’ concern for health increased from 7.6 in 2018 to 9.4 in 2020. Approximately 47.8% of the respondents considered it necessary to purchase health protection products [6]. It is foreseeable that Chinese families have a strong demand in many areas, such as elderly protection, chronic disease, special disease protection, and long-term care. The potential market for family health investment is expanding [7].

However, it is noteworthy that participation in household health investment remains low. As of 2019, only 10% of households have chosen to participate in commercial health insurance. There is also a large gap between the proportion of Chinese households spending on health care (8.8%) and developed countries (20%). Health investment, as a financial decision-making behavior with both protection and investment attributes, requires decision makers to have specific financial knowledge and risk perception ability to translate their willingness into actual participation, in addition to the influence of generalized characteristics such as age and education. Theoretically, involvement in household health investment decisions may be related to the population’s financial literacy level [8]. In the theory of financial literacy, it is an essential component of human capital, which includes not only financial knowledge and financial management skills acquired by individuals but also self-confidence in future financial planning and information processing ability [9]. These dimensions cover the essential competencies required in an individual’s investment process. This implies that individuals with low financial literacy need help making effective investment decisions or even fear making decisions [10]. Individuals with high financial literacy are more willing to make multiple risky investments and more likely to allocate a more reasonable investment portfolio [11,12]. Therefore, as the leading financial bearer of the household and the decision maker of financial asset allocation, householders will be deeply involved in the *HHI* decision-making process. The householders’ financial literacy level will be essential in the *HHI* decision-making process.

Researchers have extensively analyzed the factors influencing health investments, dating back to the Grossman model. This points out that the health need motivates individuals to invest. Individuals who spend more time being healthy can reap the corresponding health investment benefits (e.g., more work hours and income) [13]. Subsequent research builds on this model by exploring the effects of age, mood, education, occupational environment (proximity to pollution), and asset changes [14,15]. However, there needs to be more research on financial literacy and health investments. Research in financial literacy has also focused on the interrelationship between financial literacy and financial market participation. Therefore, we investigate the participation rate and decision choice of health investment based on the financial literacy perspective of household heads and empirically analyze its impact using data from the 2019 China Household Finance Survey (CHFS).

The potential contributions of this study: (1) We focus on the participation and decision-making behavior of household health investment and enrich the literature base on health investment behavior by providing additional explanations of the actual situation, influencing factors, and mechanisms of action. (2) The development of financial literacy theory has a certain lag compared with the actual situation. We choose financial literacy as an entry point to research household health investment, further enriching the study of financial well-being theory. (3) We decompose financial literacy into two levels (primary and advanced) and explore the marginal effects of its impact on household health investment, which enriches the knowledge of the realization paths related to expanding the coverage of family health investment.

The remaining parts of this study are organized as follows. The theoretical basis and research hypothesis are presented in Section 2. The methodology is shown in Section 3. The empirical results are displayed in Section 4. Section 5 aims to conclude the research, address specific policy implications, and list some research prospects. Additionally, an outline of the research framework is shown in Figure 1.

## 2. Theoretical Basis and Research Hypothesis

Three types of literature are relevant to the study in this paper. First, the literature related to financial literacy measurement. Xiao et al. [16] note that the common denominator of financial literacy tests focuses on assessing one’s knowledge base related to economic concepts and calculations. Still, this view may need to be revised. Kim and Mountain [17] assign values to responses to financial literacy questions and argue that adding “don’t know (DK)” or “reject (RF)” answers are essential, which can lead to bias in regression if DK/RF responses are not randomized. The regression will lead to bias if the DK/RF responses are not random. These studies provide the basis for this paper.

Second, the literature on financial literacy affects financial decision-making behavior. Based on investment psychology’s research framework and planned behavior theory, Raut [18], Yin, and Yang [19] point out the negative relationship between financial literacy and individuals’ social stress and psychological and cognitive bias. They argue that higher financial literacy can effectively improve individuals’ decision rationality.

Third, the literature on financial literacy affects household asset allocation. Yang et al. [20] use questionnaire data to test the positive relationship between the level of financial literacy and the number of various types of financial assets and points out that household heads with higher financial literacy are more likely to participate in specific financial markets. Lu et al. [21] point out that higher financial literacy means a higher score in asset allocation assessment because households with higher financial literacy pay more attention to financial and economic news and tend to obtain information through the information channels of investment advisors.

### 2.1. Impact of Financial Literacy on Household Health Investments

Financial literacy is the human capital required for household financial management (non-market production process) [22]. The content of *FL* includes several dimensions, such as financial knowledge, financial management skills, and decision-making skills. They are the main competencies needed in the household investment decision-making process. Thus, households with high *FL* have an advantage over other households regarding technical knowledge, decision-making ability, etc. This makes them more likely to participate in *HHI*.

First, according to financial literacy theory, investors with low financial knowledge and numeracy skills need help making scientific investment decisions [23,24]. Health investment products have complex return measurement models (involving pricing, costing, professional thresholds, and complex arithmetic). This means that numeracy skills and understanding basic financial terms can help households understand health investment products’ characteristics. Second, householders need to spend a lot of time and effort to obtain and analyze investment information to make proper decisions. Financial literacy is the basis for recognizing and understanding the contractual framework of investment transactions. This implies that *FL* is vital in screening and analyzing investment information [25,26]. Existing studies confirmed this. Household heads with high *FL* are also able to estimate product credibility as well as product return risk more accurately. They are also happy to seek help from financial service providers for professional and personalized service advice. Higher *FL* household heads are less likely to be misled by false information about health investment products [27]. Third, health investment has a protective character compared to financial products such as stocks and funds. This reduces the health risks and the resulting financial risks for household members, which is a critical way to achieve the economic well-being of households [28,29]. In family finance theory, households with high *FL* enable them to make more informed decisions about allocating resources [30,31]. Therefore, householders with high *FL* are more likely to focus on and engage in avenues of household health investments to achieve long-term financial well-being. This study proposes the following research hypotheses based on the above analysis.

**Hypothesis 1.** 
*There may be a positive relationship between financial literacy and household health investments, with higher financial literacy of household heads making them more likely to engage in household health investments.*


### 2.2. Mediating Effects of Financial Literacy on Household Health Investments

#### 2.2.1. Health Literacy

Health literacy is among the main drivers for households to engage in health investments [32] (Figure 2). It is not only people’s perception of health but also the health care knowledge and life philosophy needed to maintain their health. First, existing studies point to the adverse effects of cognitive limitations and irrational stimuli on individual health behaviors. Financial literacy as a decision-making tool is associated with higher levels of education and rational decision making. Specifically, households with higher levels of financial literacy have a higher level of cognitive ability and acceptance of health issues and health knowledge. They will have more accurate disease risk estimates and be more sensitive to the lack of health coverage capacity [33]. This discourages household members from engaging in irrational behaviors (e.g., smoking and gambling) [34] and increases healthy behaviors (e.g., daily exercise and good lifestyle habits) [35,36]. Therefore, households with higher levels of financial literacy will have a stronger motivation to invest in health. Secondly, in the process of health literacy acquisition, screening health knowledge and access to channels are essential [37]. Individuals with higher financial literacy may be more likely to screen information successfully because they rely more on formal methods (e.g., professional seminars and authoritative experts) than on non-authoritative institutions or family members, relatives, colleagues, etc. [38]. Individuals with this preference have more reliable information sources and can obtain accurate health knowledge after the reasonable screening to further improve their health literacy. Therefore, improved financial literacy will promote health literacy and make household heads correctly perceive the health care functions and effects of healthy investment products and be more willing to accept family health investments.

**Hypothesis 2a.** 
*Health literacy may be a critical mediating variable in financial literacy influencing household health investment.*


#### 2.2.2. Information Channels

According to behavioral finance theory, individuals need to go through three stages in the decision-making process: information acquisition, processing and evaluation, and decision making [39,40] (Figure 2). Whether to invest in health financial products is an economic decision of the household head after considering multiple information: firstly, the household head needs to have the ability to correctly perceive the health investment function to evaluate and process the data; secondly, the household head needs to effectively process the relevant insurance information they have obtained, accept and approve the insurance products, and thus make an effective insurance decision [41]. This process is closely related to the ability of the household head to obtain relevant financial information. Individuals with higher financial literacy usually collect relevant financial information actively and thus have more comprehensive access to health and financial news and are more likely to access health investment products [42]. At this point, the household head can not only choose whether to accept health investment products based on the financial information received and the economic environment in which they live but is also more likely to select and allocate a suitable investment portfolio. Considering the apparent information asymmetry in the health investment market, the financial literacy of household heads will have a broader impact on household health investment decisions.

**Hypothesis 2b.** 
*Information channels may be the critical mediating variables in financial literacy influencing household health investment.*


#### 2.2.3. Household Income

With the gradual improvement of financial literacy, households are more likely to accumulate wealth through channels such as increasing income and rational planning of savings (Figure 2). Firstly, the financial literacy of householders is positively correlated with their education level, which also implies that they are more likely to be in high-return industries or start businesses to obtain more wage income. They are also more likely to rationalize and lower the cost of risky investments when faced with diversified and complex financial products, thus increasing their property income [41]. Second, household heads with higher financial literacy can better understand financial institutions’ relevant systems and regulations. They can reasonably arrange household savings according to the economic development environment, rationally choose financial products from formal financial institutions, and are more likely to plan for retirement [43]. All these are conducive to the accumulation of household wealth.

Similar to consumer behavior, health investment behavior is a process by which household heads make decisions based on the maximum income they can earn over the life cycle. This has been well documented that income is an essential factor influencing individual investment behavior [44]. On the one hand, wealthier households have a higher need for risk aversion. On the other hand, after meeting the basic material needs of life, household heads have an adequate budget to participate in healthy investments with their surplus income [45]. Thus, the financial literacy of household heads can influence health investment behavior by increasing household income. Investment in health protection enhances the human health capital of household members, which leads to higher household income, forming a virtuous circle.

Based on the above analysis, this research puts forward the following research hypothesis:

**Hypothesis 2c.** 
*In financial literacy influencing household health investment, household income level may be a critical mediating variable.*


## 3. Data, Model Specification, and Variables

### 3.1. Dataset

The data used in this study are from the China Household Financial Survey (CHFS), a nationwide sampling project conducted by the China Household Financial Survey and Research Center (CHFSRC). The data from the fifth round of the survey in 2019 cover 29 provinces (regions and cities), with a sample size of 34,643 households, which is nationally and provincially representative of China.

This study focuses on financial literacy and household health investment using the latest CHFS (2019) dataset and combines household and adult questionnaires for variable selection. These data are selected for the following reasons: First, the CHFS questionnaire contains a specialized financial knowledge module with questions that conform to the international customary side measurement standards and are representative, covering three dimensions of financial knowledge, financial calculation, and financial attitude, including objective questions on interest rates, simple interest, compound interest, inflation, risk perception, and also subjective questions on formulation. Second, the questionnaire includes a module on household financial allocation to measure financial behavior better. Third, the data in the questionnaire on the individual characteristics of household heads, social features, and other related contents are also highly comprehensive and meet the needs of this study.

In collating the data, we eliminate the non-household head sample and the sample with missing financial literacy indicators. Secondly, to prevent the effects caused by extreme values, we eliminate the variables of household head age less than 20 years old and more than 80 years old. Finally, we remove some missing values and outlier samples in the control variables and obtain a valid piece of 32,090 households.

### 3.2. Variable Selection

#### 3.2.1. Financial Literacy

The core explanatory variable of interest in this paper is the level of financial literacy. We use factor analysis to construct financial literacy variables based on the questionnaire’s answers to financial knowledge and competence questions. The selected indicators include interest rate, inflation, exchange rate, financial products, whether to hold demand deposits, whether to have time deposits, and profit measurement (The definitions of the indicators are in Appendix A Table A1).

Since it is generally accepted in academia that the level of financial literacy represented by an incorrect answer differs from that described by a response that does not compute or does not know. We constructed two dummy variables of “whether to answer correctly” and “whether to answer directly” (do not know or cannot calculate as an indirect answer) and finally formed a financial literacy index system with 27 variables. Based on the principle of eigenvalues greater than or equal to 1, we retain nine public factors, including financial knowledge, attitude, and behavior. The results are shown in Table 1.

Table 2 reports the results of the reliability and validity tests of the data, which show that the KMO value is 0.813 and the *p* value of Bartlett’s sphericity test is less than 0.001, indicating that the sample is suitable for factor analysis. Based on the factor loadings of each variable in Table 2, the financial literacy indicators of this paper can be calculated using Bartlett’s method.

#### 3.2.2. Household Health Investment

According to the Grossman model, health is also a consumer good. Health investment is a consumer behavior undertaken to pursue physical and mental health through economic expenditure on purchasing and acquiring health products and services (mainly including health care investment, health care investment and auxiliary supplies, etc.). This behavior is an investment to reduce or avoid the attack of diseases on households or individuals to maintain their working capacity and is mainly realized through preventive health care measures for healthy people. Health investment aims to reduce future health expenditures, so it is necessary to distinguish health investment and health spending. Existing research indicates that health investments are payments made when the body is in relative health (health care). Health spending is compulsive behaviors undertaken to improve illness and restore health.

Therefore, we focused on two types of decisions made by householders around household health investments: First, the likelihood of household participation in health investments, measured by “whether the respondent’s household purchased commercial health insurance in the last year”. Second, investment diversity, measured by the number of health investment products purchased by the household, is divided into four categories in the questionnaire: general commercial health insurance, critical illness insurance; income protection insurance; and long-term care insurance.

#### 3.2.3. Control Variables and Mediating Variables

Household health investment behavior is a combined decision-making behavior of the household head based on individual and household factors. We select control variables for the reliability of data sources and the scientific nature of the study. First, personal characteristics. Age, gender, education, marital status, risk appetite, health status, and health literacy were selected. Second, the household characteristics. Variables related to annual household income, housing status, and household size. Then, social factors. We choose information channels and happiness in life. In particular, health literacy is more challenging to measure due to the limitations of the questionnaire data. In the health belief model, individuals’ perceptions of the likelihood and severity of illness are the drivers of health behavior, and with increased health awareness, lifestyle behaviors will improve [46]. Therefore, many scholars use health behavior as a proxy variable for health awareness [47,48]. In conjunction with the question design of CHFS (2019), we selected the question “Household consumption for preventive health behaviors (including tonic health products, blood pressure and blood glucose meters, massage and health care devices, and birth control products)”. Specifically, we set dummy variables based on the answers to this question and used the willingness of respondents’ households to pay for health behaviors as a proxy for health awareness. To better reflect the Chinese social reality, we divide the education level indicators into three dimensions of illiteracy, high school, and university for testing; we also exclude the indicators of age from the samples less than 20 years old and more than 80 years old.

The mediating variables include health literacy, information channels, and household income. Since the health literacy of the household head was not directly investigated in the questionnaire, we chose “whether or not spends on self-care” to indicate the health literacy of the householders because individuals with higher health literacy are more willing to pay for self-care. The other two mediating variables are shown in the control variables. Detailed descriptions of the main variables are shown in Table 3, and descriptive statistics are shown in Table 4.

#### 3.2.4. Variable Descriptions

Table 4 shows the descriptive statistics of the variables. In the survey sample of CHFS 2019, the participation rate of *HHI* is only 3.7%; the dispersion of the investment dispersion indicator is high (its standard deviation is 0.233), and the premium expenditure data are volatile; all of these imply that the household heads in the sample are not willing to participate in household health investment and their behavior in terms of financial decision making varies widely.

In addition, the level of financial literacy is low, with the sample’s mean being less than half of the maximum value. Most household heads hold a cautious attitude toward risk (the mean degree of risk appetite is less than half of the maximum value). They have poor access to information in financial management. Among the control variables, the average age of the sample is 56 years old (which indicates that most household heads are at an age where they can make household decisions, and it is less likely that the empirical results are affected by the inability to make essential household decisions). Only 24.3% of the sample was female; approximately 6.7% of the household heads were uneducated; 85.6% of the household heads were cared for by a partner, and the total number of people in the surveyed households averaged 3. We also found that the overall health level of the sample was good (mean score of 3.27), with most household heads having a social security account (94.1%). However, health literacy was low (only 17% were willing to spend money on health care).

### 3.3. Model Specification

To investigate the effects of financial literacy on the participation and decision-making of household health investment, this study employed two econometric methods, including the Probit model and the negative binomial model. The corresponding econometric model was established as follows.
(1)$$HHI_{i}=a_{1}+\beta_{1}FL_{i}+\gamma_{1}X_{i}+\varepsilon_{i}\;$$
where *β* refers to the total effects of financial literacy on the explained variables, *γ* refers to the coefficient of other control variables; *α* represents the individual’s effect; *HHIi* denotes household health investment; *FL_i_* denotes financial literacy, *X_i_* denotes the control variable; *i* denotes householders, and *ε_i_* is the random disturbance term.

We used two indicators to indicate the *HHI_i_* variable. These two explanatory variables are discrete and do not meet the linearity assumption of OLS, which may lead to heteroskedasticity in the results. Therefore, we used the discrete choice model to design the empirical study.

First, *HHIP_i_* denotes participation in household health investment. Equation (1) can represent financial literacy’s effect on household participation in health investments. *HHIP_i_* belongs to the choice behavior of household participation in health investments and is a binary choice variable. In the optional discrete choice model, the partial regression coefficient of the Probit regression represents the change in the probability density function of an outcome for each unit increase in the independent variable when the other independent variables are held constant. Therefore, we chose the probit model to reflect the degree of influence of household head *FL_i_* in the case of consistent personal, household, and other conditions. This will help explain the role of *FL* in the household health investment process.

Second, *HHIV_i_* denotes the type of household health investment. Equation (1) can represent financial literacy’s effect on the diversification of household health investments. Considering that household heads can choose to invest in several health care products simultaneously and that investing in these products is not mutually exclusive. In this paper, we used the number of varieties of household health investment products purchased to measure *HHIV_i_*, which is a non-negative count data. Typically, Poisson regression is used for such count data. However, the *HHIV_i_* data do not match the assumptions of Poisson regression. The expectation and variance of Poisson distribution must be equal. Instead, we found that the variance for *HHIV_i_* was 0.054, and the Expectation for *HHIV_i_* was 0.043. Therefore, we used the LR test to determine whether the *HHIV_i_* data were excessively discrete (see Appendix A Table A2). The results showed that the alpha confidence interval of the LR test was between [1.37, 2.27], which rejected the original hypothesis of the parameter “α = 0” (corresponding to Poisson distribution) at the 1% significance level. This proves that the *HHIV_i_* data obey a negative binomial distribution. The negative binomial regression model proposed by Cameron and Trivedi was chosen to test the effect of *FL_i_* on *HHIV_i_*. In addition, we used robust standard errors to improve the estimation efficiency.

This study also focuses on how financial literacy affects household health investments. Financial literacy can promote household health investment by improving information gathering, health literacy, and increasing household income. Therefore, we draw on the test steps of Methieu and Taylor [49] and add two mediating variables, household income and health literacy, to explain the issue of the mediating mechanism of financial literacy level promoting household health investment.
(2)$$M_{i}=\rho_{1}+\theta FL_{i}+\omega_{1}Z_{i}+\mu_{i}$$
(3)$$HHI_{i}=\rho_{2}+\lambda FL_{i}+\eta M_{i}+\omega_{2}Z_{i}+\mu_{i}$$

Among them, *M_i_* donates the mediating variable, including health literacy (*HL*), information channels (*IF*), and household income(*IC*); *HHI_i_* donates the explained variables that have been expressed in the Equations (1)–(3); *Z_i_* donates control variable (removing the mediating variables); *θ* refers to the effects of financial literacy on the mediating variables, *λ* refers to the direct effects of *FL_i_*, *η* donates the coefficient of the effect of mediating variable on *HHI_i_*, *ω* refers to the coefficient of other control variables, *ρ* represents the individuals effect; *μ_i_* is the random disturbance term.

The Probit and negative binomial model results only report the significance and sign of the parameters and cannot intuitively explain the economic implications. In this paper, we will calculate the average marginal effects of explanatory variables such as financial literacy on the impact of household health investment. In addition, we estimate the model using robust standard errors to eliminate the effect of heteroscedasticity due to the possible similarity of individual differences in the sample.

## 4. Results

### 4.1. Baseline Regression Results

We empirically investigate the relationship between financial literacy and household health investment using the Probit and negative binomial models (*NB*). The regression results are presented in Table 5.

Columns (1)–(4) in brackets are the coefficient of financial literacy (*FL*) on family health investment participation (*HHIP*). After adding the control variables, the average marginal effect of financial literacy changes from 0.055 to 0.016, and both coefficients are significantly positive at the 1% level. This indicates that the increase in financial literacy of the household head can increase participation in household health investment. In the binary choice model, the frequency of the explanatory variables affects the estimation bias, and the “rare event bias” still exists even if the sample size reaches thousands [50]. According to the sample description, only 3.7% of individuals have commercial health insurance, a rare event. We draw on the method of King and Zeng [51] to obtain bias-corrected estimates. The regression results are presented in Column (4), which finds that the positive effect of financial literacy on household health investment participation is still significant at the 1% confidence level.

Columns (5)–(8) tested the effect of *FL* on *HHIV*. It can be found that householders with higher financial literacy are more likely to purchase multiple commercial health insurance policies. To measure the decision response of financial literacy, we calculate the Incidence Ratio Rate (*IRR*) of *HHIV* in Column (8). The coefficient of *FL* indicates that, given other variables, each 1-unit increase in household head *FL* is associated with a 38.4% increase in the average incidence of purchasing multiple *HHI* products (implying that, with an equal endowment, householders purchase multiple health investment products is 0.384 times higher for those with higher *FL* than for those with lower *FL*).

The results of the control variables showed an inverse V-shaped relationship between the age of the household head and household health investment, which may be related to the head’s physical changes and life experiences over the life cycle. Adolescents usually have more muscular bodies, but physical function decreases as the household head ages. Chinese household heads are also typically under heavy life stress in middle age (mainly due to the deteriorating health of the household head’s parents, increased work stress, children approaching adolescence, and increased education costs for children). They are financially vulnerable, requiring the purchase of protection products to reduce financial risk. The finding that female heads of household are more likely to engage in family health investments is consistent with reality. This may be related to the more cautious nature of women. The positive effect of education level and risk appetite on household health investment is consistent with previous studies. Households with more family members are less likely to engage in family health investments. This may be related to the traditional support culture of the child support model and the clan support model. In particular, household health investments that are willing to pay for preventive health behaviors may be because they are more health conscious. The results of household income and information-gathering ability, which are important influencing factors in household health investment, show significant positive effects.

### 4.2. Regression Results Considering the Degree of Financial Literacy

Financial literacy includes both knowledge and ability. We decompose the financial literacy index into two parts: primary financial literacy at the knowledge level (knowledge factor, product awareness factor, exchange rate knowledge factor, and risk factor) and advanced financial literacy at the ability level (decision factor). The effects of the degree of *FL* on *HHI* are then examined separately, and the results are presented in Table 6. In addition, to better analyze household heads’ health investment behavior, we selected the premiums paid in the health investment process (*HHI* Cost) for analysis (considering the premium discount received by public-sector employees, we removed the sample of public-sector employees).

It can be found that, in participation in healthy household investment and product selection, the *PFL* plays a more positive role (Panel A). This implies that financial knowledge and understanding of product terms are more critical than the ability to use asset allocation and risk measurement in the investment decision-making stage. The findings provide an idea for the implementation of financial socialization. Improving household heads’ financial literacy requires attention to financial literacy and developing their financial capabilities. In addition, personalized financial socialization services need to be targeted to individuals with different knowledge bases. When promoting healthy investment products, it is essential to focus on improving the product system and the level of protection, simplifying the transaction procedures, and optimizing the readability of the terms and conditions.

### 4.3. Heterogeneity Discussion

The typical characteristics of the urban–rural dual economic structure exist in China, resulting in significant disparities between urban and rural areas regarding productivity, living standards, income levels, social security, and medical services. Thus, we divide the sample into urban and rural groups for group regressions, and the results are shown in Panel A (see Table 7). By comparison, we find that the average marginal effect is more significant in the urban group than in the rural group, which is intuitive. Urban areas have a better health care system and a higher level of medical care than rural areas, and urban areas have access to better information services, making the urban sample more inclined to pay attention to their health status and choose to invest in their health. In addition, the high income of urban group households eases the financial constraints they face. For rural residents, the level of local financial development is low, and the improvement of financial literacy may not be a good incentive for them to participate in health investments.

On the other hand, in the context of the urban–rural dual economic structure, a portion of the mobile population moves from the village to the city for work. They flow to urban areas in the form of outgoing workers. The high mobility of this population, which constantly changes its place of residence and work, also makes it very difficult to establish a social security system for this group of people. We screened this group from our sample. The results from Panel B show that the migrant population does not significantly increase their participation in health investments, even if they have a high level of *FL*. Possible reasons for this are that rural migrants are often in unstable work situations and have poor life stability. Most migrants do not have a clear understanding of social security issues. They may not have access to health investment opportunities.

In addition to the urban–rural differences, the age difference of householders is also an element that needs further analysis. Considering that China’s economic development took a significant turn in the 1970s, those born after the 1970s received a better nine-year compulsory education and experienced China’s economic takeoff and social transformation after graduation. Their different life experiences led them to form new individualistic values, emphasize self-growth, and have a higher degree of acceptance of new things. We divide the sample into pre-1970s and post-1970s groups for regression (Panel C). *FL* has a more positive effect on the *HHI* of post-1970s householders (grade up to 50 years old).

In general, older adults with more savings and a greater willingness for health literacy are more likely to participate in *HHI*. However, the results of Panel B suggest that personal acceptance of health investment products plays a more important role than their own health needs. The population of participants in *HHI* is likely to become younger and younger. Therefore, in promoting family participation in health investment, we should pay attention to the asset allocation preferences and values that individuals developed in different contexts. Employment status also needs attention as an essential household contextual characteristic. In China’s health care system, public-sector employees are reimbursed at a higher level and may receive higher premium discounts than other employees. This situation may have a differential impact on households with different work backgrounds. As seen in Panel D, public-sector employees with higher *FL* are more likely to participate in health investments or purchase multiple health investment products. Public sector jobs offer more excellent stability and sustainability and are favored by prudent and risk-averse individuals. When individuals obtain higher *FL*, they are more likely to continue to choose the certainty of coverage (such as participation in health investments) to hedge against the risks associated with health issues. This finding can provide insight into identifying target customers for health investment products.

### 4.4. Robustness Test

#### 4.4.1. Endogeneity Issues

In the regression results above, financial literacy may have endogeneity problems due to reverse causality and omitted variables. In general, financial literacy is also influenced by healthy household investments. Individuals’ participation in financial market investments can enrich investment experience, enhance risk perception, and exercise investment profitability, thus improving financial literacy. In addition, respondents may underestimate the role of financial literacy by giving inaccurate answers or relying on guesses during questionnaire responses.

Based on these two points, we draw on Bucher and Lusardi [52] to mitigate the estimation bias due to endogeneity. First, we choose the mean of community financial literacy as an instrumental variable. Most households choose properties based on commuting convenience and cost, which makes residents in the community generally have similar work backgrounds or incomes (e.g., school district housing in China and unit pool housing). Householders can improve their financial literacy by learning from their neighbors, and others’ financial literacy is strictly exogenous relative to heads of households. This makes the average level of financial agglomeration within a community more plausible in the Chinese data analysis. Second, we choose the partner’s education level as an instrumental variable for testing. Individuals are influenced by the traditional Chinese idea of “household matching” to find a partner similar to them to form a family; the partner’s education experience is also strictly exogenous to the head of the household [53]. Finally, we selected “the extent to which households pay attention to information on economic and financial aspects” as a proxy variable for FL. Generally, households more concerned about financial information will have more financial knowledge and higher financial literacy.

The results of the two-stage instrumental variable estimation are shown in Table 8. The Durbin–Wu–Hausman test shows no weak instrumental variable, and the original hypothesis that financial literacy is not endogenous is rejected at the 1% level. The first stage estimates indicate that instrumental variables have reliable explanatory power for endogenous variables. The marginal effects of financial literacy in the second stage are both significant at the 1% level (0.415, 0.048, and 0.068, respectively). This suggests that financial literacy makes an essential determinant of household health investment decisions.

#### 4.4.2. Robustness Test with Substitution of Explanatory Variables and Deletion of Samples

We also perform robustness tests on the regression results by replacing the explanatory variables and deleting the samples. In Table 9, columns (1)–(2) measure financial literacy by summing the number of correct response scores of household heads following Lusardi and Mitchell [54]. Columns (3)–(4) use the *FL* of householders in the previous survey year (2017) as a proxy (CHFS is a tracking survey, and we matched households one-to-one). Columns (5)–(6) remove the sample working in the financial sector. All the above results show that the positive effect of financial literacy on household health investment remains significant at the 1% level. Compared with the previous studies’ results, they are consistent except for the difference in the magnitude of the coefficients, indicating that the models are robust.

### 4.5. Mechanism Test

According to the above analysis, health literacy, information channels, and household income are three mediating variables. We report the regression results after controlling for these three variables in the previous article section, and the coefficients show that they all significantly contribute to *HHI*. In this section, we examine the effect of financial literacy on these three variables (see Table 10). We find that the coefficients of financial literacy are all significant at the 1% level, indicating that household heads’ financial literacy can effectively broaden their access to information and improve their health literacy. In addition, improving financial literacy also effectively increases household income, which will help alleviate investment constraints. The above results suggest that these three variables are essential mediating mechanisms.

## 5. Conclusions, Policy Suggestions, and Discussion

### 5.1. Conclusions

We examine the impact of financial literacy on household health investments. Our study focused on households’ participation and investment decisions (including the diversification and costs) in health investments. The main findings are as follows. (1) Financial literacy plays an important role in household health investments. According to the China Consumer Financial Literacy Survey and Analysis Report (2021), the financial literacy index is still low and varies widely among different groups. Extensive financial education may be able to help expand health investment coverage in the future. (2) People with high levels of financial literacy view family health investments as a good risk management tool and have a stronger health awareness. They are willing to invest in their household’s health. They are 38% more likely to diversify their investments when making investment decisions. (3) Primary financial literacy (such as financial knowledge, numeracy and risk perception) is a critical factor in increasing household participation in health investment than those with more operational experience and profitability in the investment market, which provides ideas for future financial literacy education. (4) We find that social and family backgrounds have essential moderating effects on household health investment behavior, especially the social security dilemma faced by rural migrants needs further attention. In addition, different life experiences lead to different asset allocation preferences and values, which play a critical moderating role in the financial literacy of household health investments.

### 5.2. Policy Suggestions

Based on these findings, the following adjustments are needed to accommodate residents’ growing demand for health investments.

(1) Expanding the coverage of family health investments is inseparable from the residents’ financial literacy. We should motivate residents to improve their financial literacy and encourage families to make long-term development plans to increase the likelihood of health investments. At the same time, we need to enhance consumers’ awareness of risk prevention and require financial institutions to embed financial literacy and risk communication in product sales to foster rational concepts. (2) Primary financial literacy (PFL) is more important in promoting household participation in health investments than advanced financial literacy (AFL). On the one hand, we should strengthen residents’ awareness of health investment products, such as popularizing the terminology related to health investment, so that residents can understand the primary financial knowledge related to health investment. On the other hand, we should encourage financial institutions to improve the readability of transaction terms, enhance market information transparency, and improve the health investment product system and the level of protection. (3) Advanced financial literacy has benefits in optimizing household health investment behavior. The government must promote financial socialization services in-depth and provide personalized counseling for residents with different knowledge bases and investment experiences. (4) The financial infrastructure in rural areas of China still needs to be completed, and there are noticeable regional differences compared with prosperous areas. We can actively carry out financial literacy activities by developing financial education and publicity channels and methods. For example, we can regularly organize lecture groups and go into the community to carry out financial literacy activities; we can also use online media to carry out live classes regularly. In addition, the government should actively focus on the vulnerable situation of rural migrants in terms of health security and encourage financial infrastructure for vulnerable groups to enhance their trust in health investment products. Expanding the range of financial services will improve the accessibility of financial services.

### 5.3. Discussion

Our study has some limitations that need to be improved in future studies. (1) Though this paper provides an in-depth analysis of household health investments, it has not been used to assess the impact of financial literacy on the actual benefits generated by health investment behavior due to a lack of data, nor has it considered the role of subjective financial literacy. In future research, the role played by financial literacy in promoting household health investments can be further explored by establishing a benefits evaluation index or surveys to obtain residents’ self-evaluation of financial literacy. (2) Many countries have encountered challenges in expanding health investment coverage. It would be meaningful to analyze the differences in the mediating mechanisms across countries. We can compare China with other new markets, such as India and Brazil, internationally to further test the role played by residents’ financial literacy in household health investment. (3) More complicated mechanisms may be involved in the effect of financial literacy on residents’ health, which need to be discussed in depth in the future. (4) The investment cost is also a critical investment behavior. However, the limitations of the questionnaire data do not allow us to control premium discounts in the public sector or access additional insurance information. A more rational and scientific study of this issue is necessary. In future research, we hope to address this challenge as much as possible.

## Figures and Tables

**Figure 1 ijerph-20-02229-f001:**
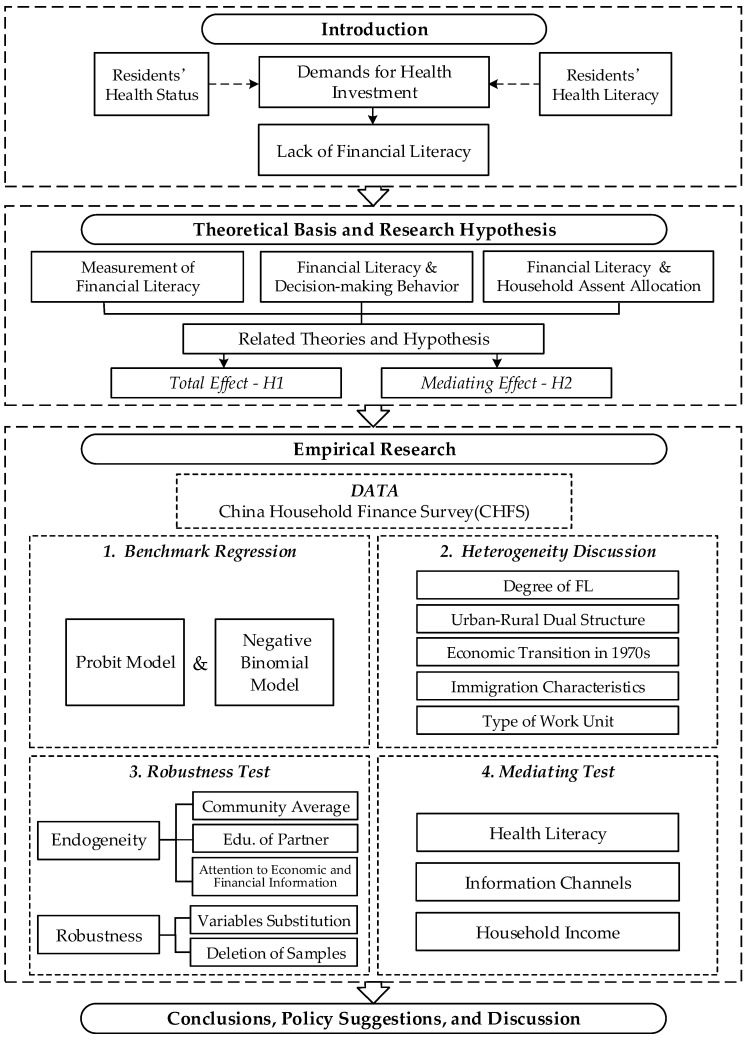
Outline of the research framework.

**Figure 2 ijerph-20-02229-f002:**
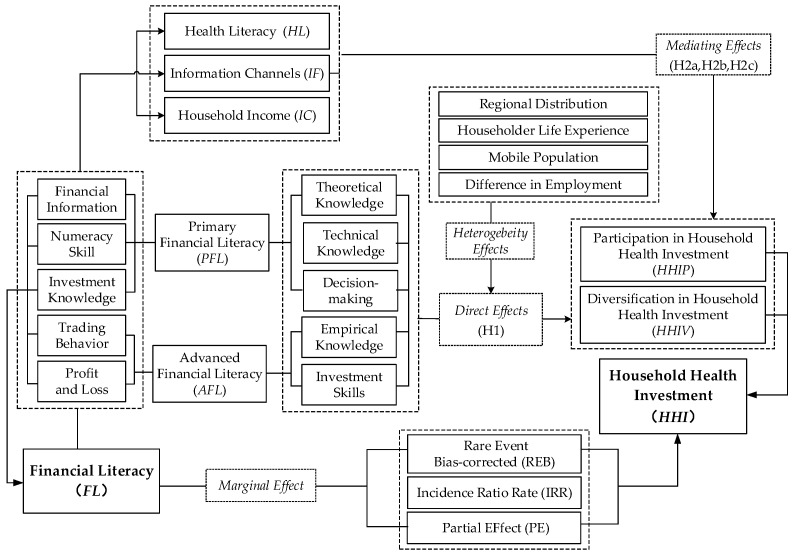
Diagram of the mechanism.

**Table 1 ijerph-20-02229-t001:** Index system and factor analysis weights of financial literacy.

Financial Literacy		Common Factor	Common Factor Weight	Index	Evaluation Criteria	Component Coefficients
Primary Financial Literacy	Financial Knowledge	Financial information	0.205	Bank prime rate for loans	Whether to answer	0.865
					True or False	0.856
		Calculation skill	0.193	Simple interest calculation	Whether to answer	0.650
					True or False	0.818
				Compound interest calculation	Whether to answer	0.601
					True or False	0.737
				Inflation	Whether to answer	0.729
					True or False	0.789
	Investment Knowledge	Risk awareness and investment understanding	0.110	Risk perception of stock	Whether to answer	0.844
					True or False	0.877
				Risk perception of fund	Whether to answer	0.779
					True or False	0.862
				Understanding level of investment	0–2 levels	0.581
Advanced Financial Literacy	Financial Behavior	Wealth investment products	0.092	Trading behavior	Whether to hold	0.888
					Whether to hold multiple	0.405
				Profit and loss	Profit or not	0.896
		Internet investment product	0.088	Trading behavior	Whether to hold	0.719
				Profit and loss	Profit or not	0.736
		Stocks	0.086	Trading behavior	Whether to hold	0.481
				Profit and loss	Profit or not	0.820
		Funds	0.086	Trading behavior	Whether to hold	0.851
				Profit and loss	Profit or not	0.879
		Other financial products	0.075	Trading behavior	Whether to hold	0.536
					Whether to hold multiple	0.782
				Profit and loss	Profit or not	0.647
		Other financial assets	0.063	Trading behavior	Whether to hold	0.853
				Profit and loss	Profit or not	0.813

Source: Summarized by the authors.

**Table 2 ijerph-20-02229-t002:** KMO and Bartlett’s test.

Bartlett’s Test of Sphericity			Kaiser-Meyer-Olkin Measure of Sampling Adequacy
Approx. Chi-Square	df	Sig.	
378,989.669	378	0.000	0.813

Source: Summarized by the authors.

**Table 3 ijerph-20-02229-t003:** Description of main variables.

Var. Name		Definitions of the Variables	Description
Participation in household health investment (*HHIP*)		Whether the participant has health insurance	Has some form of health insurance = 1; otherwise = 0
Diversification in household health investment (*HHIV*)		Dispersion of health insurance portfolio	There are 5 types of Health Insurance, the value is assigned to the number, none = 0
Financial literacy (*FL*)		Level of financial literacy	Calculate the individual scores based on factor analysis
Age (*A*)		Householder’s age	Actual age (over 20 years old and less than 80)
Female (*FG*)		Gender of householder	Female = 1, male = 0
Education	Illiteracy (*IL*)	Level of education	Illiteracy: Never went to school = 1, otherwise = 0; High: had been went to high school = 1, Otherwise = 0; College: had bachelor’s degree/master’s degree/doctoral degree = 1, otherwise = 0
	High (*HE*)		
	College (*CL*)		
Marriage (*M*)		Whether the participant is married	Married = 1, otherwise = 0
Health (*H*)		Level of householder’s health	Assign values from very poor = 1 to very good = 5
Health literacy(*HL*)		Total amount of daily self-care expenditures	Logarithm of self-care expenditures
Medicare account (*MA*)		Whether the participant has Medicare account	Has a Medicare account = 1; otherwise = 0
Income (*IC*)		Household income	Logarithm of income in the past year
Asset (*ASS*)		Household total asset	Logarithm of total asset
Happy (*HP*)		Life satisfaction	Thinks that the lifetime is generally very good = 5, very bad = 1
House (*HS*)		Number of household members	the value is assigned to the number
Family size (*FS*)		Number of family member	How many members in the family
Risk (*RS*)		Risk appetite	Assign values from risk averse = 1 to risk appetite = 5
Information (*IF*)		Access to obtain financial information	The number of channels to buy financial products; if the family has a financial advisor or investment advisor then add one to the count

Source: Summarized by the authors.

**Table 4 ijerph-20-02229-t004:** Summary statistics.

Var. Name	Obs.	Mean	Standard Deviation	Min.	Max.
*HHIP*	32,090	0.0374	0.190	0.00	1.00
*HHIV*	32,090	0.0428	0.233	0.00	4.00
*FL*	32,090	0.0007	0.364	−0.27	7.00
*A*	32,090	56.2563	12.763	20.00	80.00
*FG*	32,090	0.2428	0.429	0.00	1.00
*IL*	32,090	0.0665	0.249	0.00	1.00
*HE*	32,090	0.1937	0.395	0.00	1.00
*CL*	32,090	0.1358	0.343	0.00	1.00
*M*	32,090	0.8566	0.350	0.00	1.00
*H*	32,090	3.2698	1.002	1.00	5.00
*HL*	32,090	1.1644	2.574	0.00	12.61
*MA*	32,090	0.9409	0.236	0.00	1.00
*IC*	32,090	10.6101	1.452	−1.89	16.31
*ASS*	32,090	12.7666	1.702	0.00	21.47
*HP*	32,090	3.8486	0.862	1.00	5.00
*HS*	32,090	0.9059	0.292	0.00	1.00
*FS*	32,090	3.1334	1.539	1.00	15.00
*RS*	32,090	1.5830	0.991	1.00	5.00
*IF*	32,090	0.1098	0.403	0.00	7.00

Source: Summarized by the authors.

**Table 5 ijerph-20-02229-t005:** Benchmark regression results of financial literacy and household health investment.

Variables	*HHIP*				*HHIV*			
	(1)	(2)	(3)	(4)	(5)	(6)	(7)	(8)
	Probit	Probit	Probit	Probit (REB)	NB	NB	NB	NB (IRR)
*FL*	0.055 ***	0.017 ***	0.016 ***	0.398 ***	0.897 ***	0.017 ***	0.016 ***	0.384 ***
	(0.0030)	(0.0027)	(0.0027)	(0.0753)	(0.3218)	(0.0031)	(0.0030)	(0.0666)
*A*		−0.015 ***	0.043 ***	−0.428 ***		−0.019 ***	0.069 ***	−0.432 ***
		(0.0009)	(0.0068)	(0.0264)		(0.0013)	(0.0095)	(0.0253)
*A^2^*			−0.006 ***				−0.009 ***	
			(0.0007)				(0.0010)	
*FG*		0.015 ***	0.015 ***	0.458 ***		0.019 ***	0.019 ***	0.433 ***
		(0.0022)	(0.0023)	(0.0670)		(0.0030)	(0.0029)	(0.0652)
*IL*		−0.020 **	−0.017 *	−0.857 **		−0.047 ***	−0.042 ***	−1.050 ***
		(0.0086)	(0.0088)	(0.3419)		(0.0160)	(0.0159)	(0.3598)
*HE*		0.005 **	0.005 **	0.218 ***		0.010 ***	0.010 ***	0.231 ***
		(0.0026)	(0.0026)	(0.0828)		(0.0036)	(0.0035)	(0.0805)
*CL*		0.009 ***	0.011 ***	0.289 ***		0.014 ***	0.017 ***	0.321 ***
		(0.0028)	(0.0028)	(0.0852)		(0.0037)	(0.0036)	(0.0830)
*M*		0.004	−0.002	0.137		0.005	−0.003	0.122
		(0.0034)	(0.0034)	(0.1050)		(0.0044)	(0.0043)	(0.0997)
*H*		0.004 ***	0.004 ***	0.117 ***		0.005 ***	0.004 ***	0.106 ***
		(0.0012)	(0.0012)	(0.0367)		(0.0016)	(0.0016)	(0.0363)
*MA*		−0.002	−0.004	−0.058		−0.004	−0.006	−0.095
		(0.0045)	(0.0044)	(0.1383)		(0.0064)	(0.0061)	(0.1432)
*HL*		0.002 ***	0.003 ***	0.071 ***		0.003 ***	0.003 ***	0.063 ***
		(0.0003)	(0.0003)	(0.0099)		(0.0004)	(0.0004)	(0.0093)
*IC*		0.004 ***	0.004 ***	0.121 ***		0.006 ***	0.006 ***	0.135 ***
		(0.0011)	(0.0011)	(0.0344)		(0.0016)	(0.0015)	(0.0349)
*ASS*		0.007 ***	0.007 ***	0.242 ***		0.010 ***	0.010 ***	0.233 ***
		(0.0009)	(0.0010)	(0.0294)		(0.0013)	(0.0013)	(0.0294)
*HP*		0.000	0.001	−0.016		−0.001	0.000	−0.016
		(0.0013)	(0.0013)	(0.0393)		(0.0018)	(0.0017)	(0.0396)
*HS*		−0.010 **	−0.013 ***	−0.310 **		−0.011 **	−0.016 ***	−0.248 **
		(0.0040)	(0.0039)	(0.1208)		(0.0052)	(0.0050)	(0.1170)
*FS*		−0.002 **	−0.003 ***	−0.045 *		−0.001	−0.003 ***	−0.029
		(0.0008)	(0.0008)	(0.0242)		(0.0010)	(0.0011)	(0.0232)
*RS*		0.003 ***	0.003 ***	0.106 ***		0.005 ***	0.005 ***	0.115 ***
		(0.0010)	(0.0009)	(0.0286)		(0.0013)	(0.0012)	(0.0284)
*IF*		0.005 ***	0.005 **	0.131 **		0.006 ***	0.005 **	0.144 ***
		(0.0019)	(0.0019)	(0.0527)		(0.0022)	(0.0022)	(0.0496)
Control variables	No	Yes	Yes	Yes	No	Yes	Yes	Yes
Observations	32,100	32,100	32,100	32,100	32,100	32,100	32,100	32,100
Pseudo R^2^	0.076	0.161	0.169	-	0.070	0.147	0.156	0.147
Wald $\chi^{2}$/F test	381.900	1480.980	1282.860	-	981.850	2037.210	1790.460	2037.210
Prob > $\chi^{2}$/Prob > F	0.000	0.000	0.000	-	0.000	0.000	0.000	0.000
Log likelihood	−4732.890	−4297.262	−4256.571	-	−5230.873	−4799.246	−4746.570	−4799.246

Note: The robust standard errors are used in the estimation. *, **, and *** indicate that the significance levels of the coefficient are 10%, 5%, or 1%, respectively. Source: Calculated by the authors.

**Table 6 ijerph-20-02229-t006:** Regression results considering the degree of financial literacy.

Variables	*HHIP*	*HHIV*	
	(1)	(2)	(3)
	Probit	NB	NB (IRR)
**Panel A: Financial Literacy (Primary)**			
*PFL*	0.011 ***	0.012 ***	0.282 ***
	(0.0017)	(0.0020)	(0.0456)
Control variables	Yes	Yes	Yes
Observations	32,090	32,090	32,090
Pseudo R^2^	0.162	0.147	0.147
Wald $\chi^{2}$/F test	1497.120	2058.840	2058.840
Prob > $\chi^{2}$/Prob > F	0.000	0.000	0.000
Log likelihood	−4296.180	−4796.3935	−4796.3935
**Panel B: Financial Literacy (Advanced)**			
*AFL*	0.007 ***	0.006 ***	0.143 ***
	(0.0019)	(0.0019)	(0.0429)
Control variables	Yes	Yes	Yes
Observations	32,090	32,090	32,090
Pseudo R^2^	0.160	0.145	0.145
Wald $\chi^{2}$/F test	1985.940	1673.450	1673.450
Prob > $\chi^{2}$/Prob > F	0.000	0.000	0.000
Log likelihood	−4809.308	−4786.770	−4786.770

Note: The robust standard errors are used in the estimation. *** indicates that the significance level of the coefficient is 1%. Source: Calculated by the authors.

**Table 7 ijerph-20-02229-t007:** Heterogeneity analysis.

**Panel A: Regional Distribution**				
**Variables**	** *HHIP* **		** *HHIV* **	
	**(1)**	**(2)**	**(3)**	**(4)**
	**Urban**	**Rural**	**Urban**	**Rural**
*FL*	0.021 ***	0.016 ***	0.023 ***	0.016 **
	(0.0036)	(0.0054)	(0.0042)	(0.0064)
Control variables	Yes	Yes	Yes	Yes
Observations	20,595	11,495	20,595	11,495
Pseudo R^2^	0.141	0.116	0.127	0.106
**Panel B: Householder’s migrant status**				
**Variables**	**(7)**	**(8)**	**(9)**	**(10)**
	**Non-migrant**	**Migrant**	**Non-migrant**	**Migrant**
*FL*	0.017 ***	0.015	0.017 ***	0.021
	(0.0027)	(0.0168)	(0.0031)	(0.0000)
Control variables	Yes	Yes	Yes	Yes
Observations	31,587	503	31,587	503
Pseudo R^2^	0.162	0.391	0.147	0.350
**Panel C: Householder’s life experience**				
**Variables**	**(11)**	**(12)**	**(13)**	**(14)**
	**Post-1970s**	**Pre-1970s**	**Post-1970s**	**Pre-1970s**
*FL*	0.027 ***	0.011 ***	0.029 ***	0.010 ***
	(0.0062)	(0.0028)	(0.0073)	(0.0029)
Control variables	Yes	Yes	Yes	Yes
Observations	10,494	21,596	10,494	21,596
Pseudo R^2^	0.110	0.138	0.095	0.134
**Panel D: Householder’s employment status**				
**Variables**	**(15)**	**(16)**	**(17)**	**(18)**
	**Public sector**	**other**	**Public sector**	**other**
*FL*	0.026 ***	0.016 ***	0.025 ***	0.018 ***
	(0.0085)	(0.0030)	(0.0087)	(0.0035)
Control variables	Yes	Yes	Yes	Yes
Observations	4130	27,920	4170	27,920
Pseudo R^2^	0.097	0.168	0.090	0.155

Note: The robust standard errors are used in the estimation. **, and *** indicate that the significance levels of the coefficient are 5%, or 1%, respectively. Source: Calculated by the authors.

**Table 8 ijerph-20-02229-t008:** Endogeneity problem solving.

Variables	*HHIP*	*HHIV*	*HHIP*	*HHIV*	*HHIP*	*HHIV*
	(1)	(2)	(3)	(4)	(5)	(6)
*FL*	0.883 ***	0.820 ***	0.819 ***	0.856 ***	1.172 ***	1.152 **
	(4.800)	(4.390)	(3.810)	(3.960)	(5.250)	(5.120)
First stage	0.415 ***					
*(FL_community = FL*)	(45.970)					
First stage			0.048 ***			
(*EDU_partner = FL*)			(42.020)			
First stage					0.068 ***	
*(FL_attention = FL*)					(44.070)	
Control variables	Yes	Yes	Yes	Yes	Yes	Yes
Observations	32,090	32,090	32,090	32,090	32,090	32,090
Wald test of exogeneity	9.430 ***	7.300 ***	7.120 ***	8.030 ***	17.500 ***	16.53 ***
Test of weak instrument	21.550 ***	18.190 ***	14.600 ***	15.770 ***	27.840 ***	26.480 ***

Note: The robust standard errors are used in the estimation. **, and *** indicate that the significance levels of the coefficient are 5%, or 1%, respectively. Columns (1)–(2) in brackets tested the mean of community financial literacy as an instrumental variable; Columns (3)–(4) in brackets tested the partner’s education as an instrumental variable. Source: Calculated by the authors; Columns (5)–(6) in brackets tested the level of attention on economic and financial aspects as the instrumental variable. Source: Calculated by the authors.

**Table 9 ijerph-20-02229-t009:** Robustness test.

Variables	*HHIP*	*HHIV*	*HHIP*	*HHIV*	*HHIP*	*HHIV*
	(1)	(2)	(3)	(4)	(5)	(6)
*FL*	0.002 ***	0.003 ***	0.017 ***	0.020 ***	0.016 ***	0.017 ***
	(0.0003)	(0.0003)	(0.0061)	(0.0070)	(0.0027)	(0.0031)
Control variables	Yes	Yes	Yes	Yes	Yes	Yes
Observations	32,090	32,090	32,090	32,090	31,833	31,833
Pseudo R^2^	0.164	0.149	0.179	0.163	0.157	0.144
Wald $\chi^{2}$/F test	1523.240	2108.730	537.170	766.110	1388.980	1900.700
Prob > $\chi^{2}$/Prob > F	0.000	0.000	0.000	0.000	0.000	0.000
Log likelihood	−4283.211	−4785.481	−1386.127	−1555.794	−4164.079	−4612.064

Note: The robust standard errors are used in the estimation. *** indicate that the significance level of the coefficient is 1%. Source: Calculated by the authors.

**Table 10 ijerph-20-02229-t010:** Mediating effects of financial literacy on household health investment.

Variables	(1)	(2)	(3)	(4)	(5)	(6)
	*HL*	*HHIP*	*IF*	*HHIP*	*IC*	*HHIP*
*FL*	0.951 ***	0.017 ***	0.879 ***	0.017 ***	0.459 ***	0.017 ***
	(0.288)	(0.003)	(0.072)	(0.003)	(0.028)	(0.003)
*HL*		0.002 ***				
		(0.001)				
*IF*				0.005 ***		
				(0.002)		
*IC*						0.004 ***
						(0.001)
Control variables	Yes	Yes	Yes	Yes	Yes	Yes
Observations	32,100	32,100	32,100	32,100	32,100	32,100
Pseudo R^2^	0.016	0.165	0.215	0.165	0.086	0.165
Wald $\chi^{2}$/F test	104.300	1265.190	4608.800	1265.190	734.980	1265.190
Prob > $\chi^{2}$/Prob > F	0.000	0.000	0.000	0.000	0.000	0.000
Log likelihood	−74,657.530	−4281.076	−8875.014	−4281.076	−52,595.154	−4281.076

Note: The robust standard errors are used in the estimation. *** indicate that the significance level of the coefficient is 1%. Source: Calculated by the authors.

## Data Availability

The data used to support the findings of this study are available from the corresponding author upon request.

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
