# Peer review of "The Impact of Financial Literacy on Household Health Investment: Empirical Evidence from China"

_ijerph, 2023, doi:10.3390/ijerph20032229_

Round 1
Reviewer 1 Report
This paper uses 2019 CHFS microdata to study the association between financial literacy and household health investment.
Results show a positive association, even after controlling for various factors.
I really like the original idea of this paper, since household health is an important topic of research. Due to the rapid economic growth of China, Chinese citizens have substantially improved their life expectancy and household expenditure on health. The cost of health care has also increased as the marketization continues.
However, I have many questions and concerns:
1. I am not sure why the authors have used three different types of models: Probit, Negative Binomial and Tobit. I would recommend the authors to use the most appropriate one for each of the situations.
2. I am not sure why the authors have not controlled for the hukou or migrant status, for assets, or for the employment status. They are important factors in health investment. Rural migrants may not have access to health investment opportunities, while public sector employees in urban China may receive discounts on their insurance premium and lead to a lower level of health spending.
3. Health spending is different from health investment. "Willingness to participate in health investments” is different from the amount spent on health. It is critical to differentiate the two properly.
4. I am not convinced that Risk Perception should be part of Primary Financial Literacy. Rick perception is the subjective judgement that people make about the characteristics and severity of a risk. It has nothing to do with literacy in CHFS.
5. I am not persuaded that stock ownership should be part in Advanced Financial Literacy.
6. Without considering the premium discount received by public sector employees, I do not think that the costs of household health investment (HHIC) should be an independent variable in this study.
Here are some minor issues:
1. The authors argue that “The health problems of Chinese residents are becoming increasingly prominent…” using the evidence that “… more than 88.5% of deaths in China are caused by chronic diseases…” I do not see the implied causal relationship.
2. "...are willing to pay 16% more in investment costs than household heads with lower levels of financial literacy." is unclear. Investment costs should be avoided, if possible.
Author Response
Replies to Reviewer #1:
This paper uses 2019 CHFS microdata to study the association between financial literacy and household health investment.
Results show a positive association, even after controlling for various factors.
I really like the original idea of this paper, since household health is an important topic of research. Due to the rapid economic growth of China, Chinese citizens have substantially improved their life expectancy and household expenditure on health. The cost of health care has also increased as the marketization continues.
However, I have many questions and concerns:
Response:
Thank you very much for your careful review and positive comments! All of your comments are very important and helpful for us to revise and improve our manuscript. According to your comments, we have made relevant changes to the whole manuscript. Please check the revised manuscript. The detailed revisions are listed below, and we respond to your comments point by point.
Comments 1:
- I am not sure why the authors have used three different types of models: Probit, Negative Binomial and Tobit. I would recommend the authors to use the most appropriate one for each of the situations.
Response:
Thank you for pointing this out and providing kind advice. In the manuscript, the explanatory variables are divided into household health investment participation (HHIP), the diversification of household health investment products (HHIV), and investment cost (HHIC). They are binary choice variables, count variables, and continuous variables. So, we selected three models as required by the type of data.
We apologize for the lack of detail in explaining why the model was chosen in our manuscript. Based on your kind suggestion, we have re-evaluated the reasonableness of the model. We have considered the data characteristics and the explanatory power of the model. We finally chose the Probit and Negative Binomial models for the empirical analysis. Therefore, we provide more detailed explanations in the model design section (see Section 3.3), including the data characteristics of the explanatory variables and the model's explanatory power. We also use the rational tests to justify the choice of the model. In addition, in conjunction with your comment 6, health investment cost (HHIC) is no longer included as an independent variable in the revision.
Comments 2:
- I am not sure why the authors have not controlled for the hukou or migrant status, for assets, or for the employment status. They are important factors in health investment. Rural migrants may not have access to health investment opportunities, while public sector employees in urban China may receive discounts on their insurance premium and lead to a lower level of health spending.
Response:
Thank you for the valuable suggestions. We add the total family asset as a control variable in the empirical analysis (see Table 5). On the aspects of migration status and employment status, we discuss them in the heterogeneity analysis section (see Section 4.3 and Table 7). These are the results:Rural migrants, as a socially disadvantaged group, are hardly involved in household health investments even if they have high financial literacy. Public sector employees with higher financial literacy are more likely to invest in family health than other employees. These findings are consistent with reality, and we hope that we will be able to analyze these issues in depth in future studies.
Comments 3:
- Health spending is different from health investment. "Willingness to participate in health investments” is different from the amount spent on health. It is critical to differentiate the two properly.
Response:
Thank you for pointing this out. Health investment has a preventive character, and investment in health is made to reduce future expenditures when health problems occur. After reorganizing the article, we found that we failed to distinguish well between these two concepts, and there was a problem with the unclear presentation. So, we described the concept of health investment in the section on variable selection (see Section 3.2.2) and identified the relationship between health investment and health expenditures. In addition, we have revised possible presentation problems of the manuscript.
Comments 4:
- I am not convinced that Risk Perception should be part of Primary Financial Literacy. Rick perception is the subjective judgement that people make about the characteristics and severity of a risk. It has nothing to do with literacy in CHFS.
Response:
Thank you for pointing this out. We wanted to use "Risk Perception" to describe an individual's perception of investment products. In the indicator selection process, we used questions from the CHFS questionnaire to compare the risk of stock and fund products. These questions differ from subjective judgments of personal risk preferences. They are objective judgments of the risk of investment products (e.g., which is riskier, a debt-oriented fund or a stock-oriented fund?). However, "Risk Perception" may easily lead to misunderstandings. Therefore, we have changed this concept to "Investment Knowledge" in our revision (see Fig.2 and Table 1). Because evaluating the risk perception ability is a common way of financial literacy measurement. To be prudent, we screened risk-related questions and re-measured financial literacy variables.
As we re-measured the explanatory variables, the coefficients in the regression results table were all changed but did not affect the conclusions (see Table 5-Table 10).
Comments 5:
- I am not persuaded that stock ownership should be part in Advanced Financial Literacy.
Response:
Thank you for the useful suggestions. We apologize for the unclear presentation. We wanted to measure whether an individual has investment experience but incorrectly chose the one-sided term "Stock Ownership." In the revision, we have changed this concept to "Trading Behavior" (see Fig. 2) and have revised the expression in Table 1.
Comments 6:
- Without considering the premium discount received by public sector employees, I do not think that the costs of household health investment (HHIC) should be an independent variable in this study.
Response:
Thank you for the useful suggestions and your point is well-taken. Due to the questionnaire data limitations, we could not control premium discounts in the public sector. Therefore, we accept your comment and no longer consider HHIC an independent variable for the study. (Table 3-Table 10 have been changed). In addition, we examined the sample of employees from other sectors (the sample of public sector employees was removed). This test is conducted using advanced and basic financial literacy (see Section 4.2 and Table 6). We find that the coefficient of the effect of advanced financial literacy on higher investment costs is half less than basic financial literacy (55.65%). We then summarize the findings in the final section and discuss the shortcomings of this part of the study (see Section 5.1-Section 5.3).
Comments 7:
Here are some minor issues:
- The authors argue that “The health problems of Chinese residents are becoming increasingly prominent…” using the evidence that “… more than 88.5% of deaths in China are caused by chronic diseases…” I do not see the implied causal relationship.
Response:
Thank you for pointing this out. We reworked the presentation of the introduction and the empirical evidence. The causal relationship between residents' health issues and empirical evidence was highlighted. Specifically, we added or modified the following (see Section 1).
“The health problems of Chinese residents are becoming increasingly prominent due to various factors such as urbanization, food structure, and climate change. Chronic diseases such as hypertension and diabetes are increasingly seen in young people. "Geri-artic diseases" are gradually showing a trend of lower age, the age structure of patients is changing, and the growth rate of chronic disease groups remains high, which brings a substantial economic burden to the healthy life of residents and national healthcare. In 2020, the prevalence of chronic diseases in China's physically examined population reached 24%, and the number of deaths caused by chronic diseases accounted for 88. 5% of annual deaths in China also caused more than 90% of the national economic disease burden.”
Comments 8:
- "...are willing to pay 16% more in investment costs than household heads with lower levels of financial literacy." is unclear. Investment costs should be avoided, if possible.
Response:
Thank you for your comments and your point is well-taken. The empirical findings suggest that financial literacy does not reduce the cost of investment, which is not fit with common sense. In order to explain this, we conducted an empirical test from primary and advanced financial literacy perspectives. We have excluded the sample in conjunction with your comments on the investment cost (comment 6). As a result, we have added or revised the following content (see Section 4.2):
“In the payout cost stage, column (4) shows that both AFL and PFL significantly increase the HHI Cost. Column (5), which solves for the Partial Effect (PE) of the left truncated variable, further illustrates the positive relationship between financial literacy and investment costs. In the sample of households purchasing health investment products (with actual payments greater than 0), each 1-unit increase in PFL causes a 7.2% increase in the cost of investment. In contrast, a one-unit increase in AFL causes a 6.6% increase in investment costs. It may be related to the way the fees of health investment products are calculated. Typically, higher premiums mean longer years of participation, and people who have higher financial literacy may be more likely to avoid "nearsightedness" and plan for healthier investments in the long term, resulting in higher investment costs.”
We have also completed the following revisions as suggested:
- We combed through the references to ensure that all cited references were closely linked to the study.
- We optimized the study design according to the suggestions. Moreover, we explained the study methods based on examining rationality.
- We revised the study's results according to the optimized study design. In order to better present the results, we optimized the design of the results table and provided detailed explanations and reasonable expansions of the results.
Thank you very much for your valuable and constructive recommendations and suggestions. We have gained a lot of new knowledge from your comments. Your suggestions provide new insight for our study. Thank you very much indeed!
Reviewer 2 Report
The theme is very interesting, keeping in mind the increasing importance of the Chinese economy.
The index table can be made more precise. The definitions could be given in the appendix.
Author Response
Replies to Reviewer #2:
The theme is very interesting, keeping in mind the increasing importance of the Chinese economy.
Response:
Thank you very much for your careful review and positive comments! All of your comments have been very important and helpful for us to revise and improve our manuscript. According to your comments, we have made relevant changes to the whole manuscript. Please check the revised manuscript. The detailed revisions are listed below, and we respond to your comments point by point.
Comments:
The index table can be made more precise. The definitions could be given in the appendix.
Response:
Thank you for your useful comments and your point is well-taken. To make the table of indicators more precise, we reorganized the calculation logic of the factor analysis method. In the table, we showed the categories of FL, FL indicator system common factors, FL-specific indicators, and the selection criteria of indicators in detail (see Table 1). At the same time, we show in the table the weights of the common factors and the component coefficients of each indicator. Based on the kind suggestion, we have removed the definition section from the indicator table and placed it in the table at the end of the manuscript (see Appendix A.1).
Reviewer 3 Report
The article deals with a very broad topic, which is confirmed by a wide selection of scientific and professional articles, but narrows it down through structured frameworks the authors ultimately give the article a scientific dimension. The methods are well explained and also well applied. Perhaps outside of China, the topic would have a different aspect depending on the health system of a particular country, but presented in this way corresponds to one country. The authors were well aware of the limitations they pointed out.
Author Response
Replies to Reviewer #3:
The article deals with a very broad topic, which is confirmed by a wide selection of scientific and professional articles, but narrows it down through structured frameworks the authors ultimately give the article a scientific dimension. The methods are well explained and also well applied. Perhaps outside of China, the topic would have a different aspect depending on the health system of a particular country, but presented in this way corresponds to one country. The authors were well aware of the limitations they pointed out.
Response:
Thank you very much for your careful review and positive comments. All of your comments have been very important and helpful for us. Thank you very much indeed!
Round 2
Reviewer 1 Report
Financial literacy can play an important role in household health investment, especially given that China is increasingly market-oriented and consumers will have to make appropriate decisions. Because of economic growth and health care investments, the Chinese have substantially improved their health outcomes and encountered more aging related issues.
Financial literacy enables households to make informed decisions about how to allocate their resources, including investments in health. Households with higher levels of financial literacy should be more likely to invest in preventive health measures, such as purchasing health insurance.
Households with higher levels of financial literacy may be more likely to plan for unexpected health expenses and make informed decisions about the cost and quality of health services.
Meanwhile, households with low financial literacy may struggle to make informed decisions about health investments. They may also be more likely to delay or forego necessary health care, which can lead to worse health outcomes and higher costs in the long run.
Overall, financial literacy is expected to play an important role in household health investment, which seems to be confirmed in this study.
The manuscript has significantly improved. However, there are still important issues:
1. Figure 1. Outline of the research framework seems to have too much information.
2. The language must be polished.
3. Since this research is largely based on cross-sectional data, the authors should not be too confident about the causal relationship. For instance, Hypothesis 1a seems too strong about the implied causal direction.
4. Hypothesis 1b needs to be rewritten.
5. Hypothesis 1c. seems irrelevant to the topic.
6. How is risk attitude considered in the financial literacy?
7. Eq.2 seems irrelevant.
8. "Primary financial literacy is more important in popularizing health investment products." What is the reference group?
9. I am unclear why the Negative Binomial model is used. Does the dependent variable—HHIV—follow the negative binomial?
10. Do people with a higher level of financial literacy pay more for HHI cost, as shown in Table 6? I am not convinced that paying more for HHI is a sign of better financial literacy.
11. It is important to improve the readability of the manuscript. "...the low level of financial literacy among Chinese residents..." What is the reference group for this statement?
Author Response
Dear distinguished reviewer,
Thank you very much for taking the time to revise and submit the manuscript. We appreciate all the comments and suggestions! We have tried our best to improve and make some manuscript changes. Please find our itemized responses below—and revisions/corrections in the re-submitted files.
We sincerely hope that we have addressed all your concerns in the revision. If our revision is not up to your standard yet, we would be happy to revise it again.
all co-authors
